ecology, evolution

Bergmann's rule, countergradient variation, *Gambusia*, genetic compensation, temperature-size rule, thermal adaptation

**Author for correspondence:**
David C. Fryxell
e-mail: dfry901@aucklanduni.ac.nz

†Co-senior authorship.

# Recent warming reduces the reproductive advantage of large size and contributes to evolutionary downsizing in nature

David C. Fryxell[1,2], Alexander N. Hoover[2], Daniel A. Alvarez[2], Finn J. Arnesen[2], Javiera N. Benavente[1], Emma R. Moffett[1], Michael T. Kinnison[3], Kevin S. Simon[1,†] and Eric P. Palkovacs[2,†]

[1]School of Environment, University of Auckland, Auckland 1010, New Zealand
[2]Department of Ecology and Evolutionary Biology, University of California, Santa Cruz 95060, CA, USA
[3]School of Biology and Ecology, University of Maine, Orono, ME, USA

DCF, 0000-0003-4543-4809; ERM, 0000-0001-9891-6217

Body size is a key functional trait that is predicted to decline under warming. Warming is known to cause size declines via phenotypic plasticity, but evolutionary responses of body size to warming are poorly understood. To test for warming-induced evolutionary responses of body size and growth rates, we used populations of mosquitofish (*Gambusia affinis*) recently established (less than 100 years) from a common source across a strong thermal gradient (19–33°C) created by geothermal springs. Each spring is remarkably stable in temperature and is virtually closed to gene flow from other thermal environments. Field surveys show that with increasing site temperature, body size distributions become smaller and the reproductive advantage of larger body size decreases. After common rearing to reveal recently evolved trait differences, warmer-source populations expressed slowed juvenile growth rates and increased reproductive effort at small sizes. These results are consistent with an adaptive basis of the plastic temperature–size rule, and they suggest that temperature itself can drive the evolution of countergradient variation in growth rates. The rapid evolution of reduced juvenile growth rates and greater reproduction at a small size should contribute to substantial body downsizing in populations, with implications for population dynamics and for ecosystems in a warming world.

## 1. Introduction

Body size is a key functional trait, dictating energy demand, prey preferences, and the overall ecological role of animals [1–6]. The accurate prediction of future ecological changes may thus depend heavily on a mechanistic understanding of how key functional traits like body size are affected by environmental changes [7–9]. Today, the most common environmental change is an increase in average local temperatures [10]. In the majority of ectotherms, higher rearing temperatures are associated with faster growth rates and smaller adult body sizes [11–14]. As such, the plastic 'temperature-size rule' of fast growth but reduced adult body size has become incorporated into models predicting the population and community outcomes of warming [15–18]. While this plastic response is often assumed to be adaptive [19,20], the role of evolution in modifying this response is unknown over the short time scales relevant to current warming.

Evolutionary responses to temperature are poorly understood because confounding factors are common along thermal gradients in nature. Latitudinal and elevational temperature gradients are commonly confounded with other putative selective agents like precipitation, resource availability, biogeography

and seasonality [21–23]. Experimental evolution studies can overcome this issue by isolating the effects of temperature as an agent of selection. However, these experiments have been restricted to simplified, controlled laboratory environments (but see [24–26]), and they have been limited to testing for evolutionary responses to temperature in smaller-sized taxa (e.g. plankton, *Drosophila*, etc. [27–33]). Thus, surprisingly little is known of how warming *per se* may cause the evolution of ecologically important traits like body size and growth rates for larger taxa in nature.

Animals in aquatic environments display the strongest and most consistent temperature–size responses [13,34], indicating there may be a common adaptive explanation for this pattern. Warmer water temperatures are likely to increase oxygen limitation and may also increase resource limitation [19,35]. Large individuals have higher overall metabolic demands [5], so they may be most challenged by this decrease in availability. If large individuals are more strongly challenged by warming, life-history theory can provide simple explanations for the evolution of reduced body size. For example, if oxygen or resource limitation increases mortality rates for large size individuals, then life-history theory predicts the evolution of earlier and greater reproduction at a smaller size [36]. Relatedly, if oxygen or resource limitation stresses large size individuals, reducing the fecundity advantage of large size, life-history theory also predicts the evolution of earlier and greater reproduction at a smaller size [36,37]. Thus, if warming alters mortality or fecundity selection in a manner that disfavours large individuals, then evolution may contribute to the expression of earlier and greater reproduction at a smaller size.

Adaptive or not, the temperature–size rule describes the pattern of warming-induced size reduction at a given life stage (e.g. parturition, maturity) via plasticity, but warming could also affect body sizes through the evolution of body growth rates. For example, warming could cause the evolution of reduced somatic growth rates after maturity as a simple by-product of greater reproductive investment [38–40]. Warming may also cause the evolution of reduced growth rates before maturity, counteracting plastic growth acceleration at that life stage. For example, populations of a variety of fish taxa show evolved increases in juvenile growth rates at higher latitudes compared with populations of the same species from lower latitudes, a pattern that counteracts plasticity and promotes growth rate similarity across environments [41–47]. However, this 'countergradient' pattern in growth is thought to emerge along latitudinal gradients due to variation in the length of the growing season, with higher-latitude populations evolving fast growth to overcome the shorter length of the growing season [23,48]. Therefore, it is less clear that increased temperature *per se* will drive the evolution of reduced growth rates under current warming.

Here, we sought to test the hypothesis that increased temperature disfavours large body size, causing the rapid evolution of reduced somatic growth rates and a shift in allocation towards greater reproduction at a smaller body size. We used populations of western mosquitofish (*Gambusia affinis*) recently established across a unique set of geothermal springs. Geothermal springs can provide useful thermal gradients that break confounding patterns found along other natural thermal gradients such as latitude and altitude [49]. Past work in this study system suggests that mortality rates

may be higher for larger individuals at warmer temperatures. Populations from warmer sites tend to have smaller body size distributions [50], despite that individual growth rates tend to increase over this range in temperatures [51,52]. Moreover, field routine metabolic rate measurements suggest that the mass-specific metabolic advantage of large size is lost at higher temperatures; metabolic scaling coefficients shift from the metabolic theory expectation of approximately 0.75 at cooler temperatures towards approximately 1 at higher temperatures [50].

We tested four predictions about selection and evolution of body size and growth in warmed environments. First, we tested whether warming alters fecundity selection to favour greater reproduction at a smaller size. To do so, we used a trait survey of wild-caught female fish, to test whether higher temperatures were associated with a decrease in the fecundity advantage of large body size. Second, using first laboratory generation (F1) adult females reared in a common environment, we tested the prediction that warmer-source populations have recently evolved an increase in reproductive effort at small sizes. Third, we measured the embryo size of these mothers to test the prediction that evolution contributes to a reduction in offspring size under warming. Fourth, we used second laboratory generation (F2) juveniles reared across five temperatures to test the prediction that warmer-source temperatures were associated with the recent evolution of reduced somatic growth rates. We expected growth rates to increase with rearing temperature due to plasticity alone, so this pattern of evolution opposing plasticity would demonstrate the recent evolution of countergradient variation in growth.

## 2. Methods

### (a) Study system

The western mosquitofish is a small (less than 6 cm), sexually dimorphic, livebearing fish that was introduced across the globe throughout the twentieth century [52] (figure 1a). Male mosquitofish virtually cease growth at maturity, while female mosquitofish, like both sexes in many fishes, exhibit indeterminate growth [53]. Mosquitofish exhibit a thermal niche common of temperate ectotherm species; they can tolerate temperatures from near-zero to approximately 40°C but require warmer minimum temperatures for reproduction (approx. 16°C [52]). Here, we study populations spanning most of this reproductive thermal range.

Mosquitofish were introduced into a single site in California (CA) in 1922 from 1 to 2 sources in Texas [54]. Mosquitofish were then spread widely within the state but documentation of their timeline and introduction pathway into specific sites is rare. Today, mosquitofish occupy geothermal springs in Inyo and Mono counties (figure 1b). The focal springs studied here are unique in that they are dammed near the spring source (electronic supplementary material, table S1). Consequently, the populations of mosquitofish in these springs comprise individuals experiencing a consistent and highly constrained thermal regime, with virtually no gene flow from other environments. Each spring has a different mean temperature, and the springs exist in close proximity, relatively evenly placed along a gradient from 18.8 to 33.3°C (figure 1; electronic supplementary material, table S1). We monitored water temperature at the focal sites for several months over a period of significant change in air temperatures to confirm that these sites were thermally stable (figure 1c).

(a)

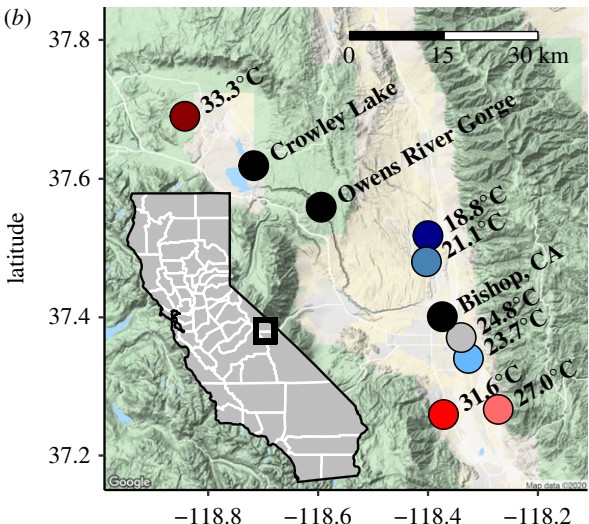

(b)

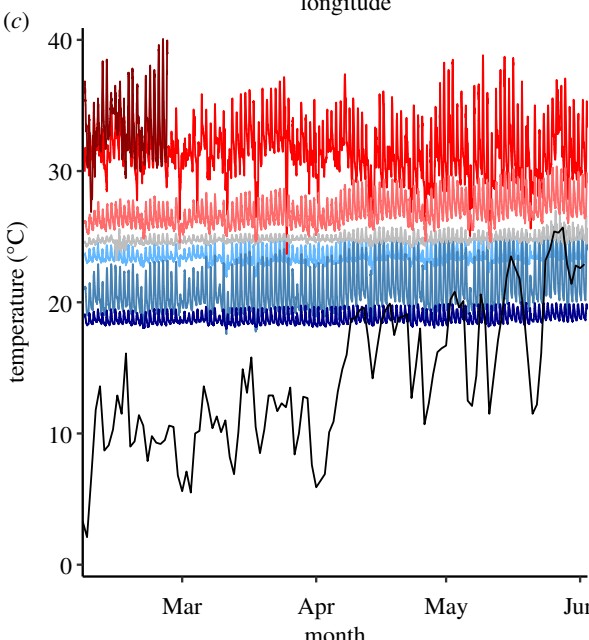

(c)

**Figure 1.** Features of the study system. (a) One of the seven geothermal spring ponds where mosquitofish were collected for this study ('AW,' 23.7°C). Inlayed is a photograph of mosquitofish in this pond. (b) A map of the geothermal spring sites (Inyo and Mono counties, California). Coloured points are sampling sites, and colours correspond to the temperature gradient. Black points are landmarks. (c) Temperature profile of each geothermal spring, logged at 15 min intervals over spring 2014. For reference, the daily average air temperature (Bishop Airport, Bishop, CA) is plotted in black. The temperature logger at the warmest site ('LHC', 33.3°C) failed at midday on 25 February. (Online version in colour.)

## (b) Fish collections

For trait analyses on wild-caught fish, we collected female mosquitofish using seine and hand nets. We sampled in spring and in summer to coarsely assess seasonality effects (see electronic supplementary material, table S2 for sample sizes and dates). Fish were euthanized with an anaesthetic overdose, immediately preserved in 95% ethanol, and later stored in 80% ethanol.

For common-garden rearing, we collected wild fish from six of the seven populations on 18 February 2018 and transported them to the University of California (UC) Santa Cruz. One site—'LAW' (24.8°C, electronic supplementary material, table S1)—had been invaded by several predatory largemouth bass (*Micropterus salmoides*) in the period between field collections and before collections for common rearing (D.C.F. 2015, personal observation), so we did not use that site for springtime sampling or common rearing. Other sites do not contain large piscivorous fishes (electronic supplementary material, table S1).

## (c) F0 and F1 rearing

Fish rearing for the wild-caught and first laboratory-born generations (F1) took place in an environment-controlled greenhouse at UC Santa Cruz (electronic supplementary material, figure S1). We introduced wild-caught adult fish to 568 l tanks (n = 6 tanks, 1 per population; electronic supplementary material, figure S1) at a density of 40 ± 5 individuals per tank. Tanks were identical and tank assignment was randomized. Tank water was off-gassed Santa Cruz city water. Each tank was heated to 26°C with a 500-watt submersible heater and water temperature was homogenized in each tank by continuous, vigorous bubbling with an air pump. Photoperiod was set to 14 : 10 h daylight : dark using full spectrum lighting. Fish were fed ground Tetramin (Tetra Holding, Blacksburg, VA, USA) flake food in morning and evening and Frystartr (Skretting Inc, Stavanger, Norway) midday. Water quality was maintained through siphoning of waste and 50% water changes twice weekly.

Newborn offspring were collected on floating fry retention devices that reduce cannibalism by adults (electronic supplementary material, figure S1). Experimental offspring were collected twice daily and retained for F1 rearing starting 18 March 2018. We waited this one-month period from adult collection to fry collection to ensure the offspring we collected were not directly exposed to their parent's natal thermal environment during early internal development. The interbrood interval of mosquitofish is about 20 days at 30°C [51]. All newborn fish from the same population born on the same day were reared together in a 'fry basket' hung in 57 l tanks in the same room and also set to 26°C (electronic supplementary material, figure S1). By 15 April 2018 we had collected at least 90 F1 fish from each population, representing estimated genetic contributions of at least 12 females per population, but probably many more (electronic supplementary material, table S3). At that point, F0 fish were euthanized, their tanks were drained, cleaned and reset, and F1 fish were introduced. F1 fish were haphazardly reduced in density to 72 ± 6 individuals for each tank. Additional F1 fish not transferred to 568 l tanks were reared in the 57 l tanks until 4 July 2018, when they were euthanized and preserved as above. On 16 June 2018, we began collection of second laboratory generation (F2) offspring as for F1 fish above. For F2 trait assays, we collected up to 10 individuals born per population per day. We continued to collect F2 fish until December 2018, at which point F1 fish were euthanized and preserved as above.

We dissected wild and F1 female mosquitofish to obtain four life-history traits related to reproduction and offspring size: reproductive allocation (gonadosomatic index, hereafter 'GSI', calculated as gonad weight ÷ total weight), fecundity (number of embryos), mean embryo diameter and mean embryo mass (F1 fish only). We sought to obtain trait data from females

across a similar range in body size from each site. We also measured embryo stage as a potential covariate affecting these traits, using a modified protocol from [55] in which one of six development stages was assigned to each brood. Our modified trait measurement protocol is provided in electronic supplementary material, appendix S1.

## (d) Adult wild and F1 trait analyses

We tested for effects of source temperature and maternal body length on the focal adult traits. We excluded non-gravid females, which were rare, from the dataset. We assigned each fish a source temperature which was the average from the time series of the population's source temperature (figure 1; electronic supplementary material, table S1). We used generalized linear models for each trait by dataset (wild in spring, wild in summer, F1) combination. We sought to remove (control for) the independent covariate effect of embryo stage, which we expected could influence life-history traits (e.g. since embryo diameter increased with embryo stage; electronic supplementary material, figure S4). We started with the full model specification: $trait \sim maternal\ length \times source\ temperature + embryo\ stage$. When the interaction was not significant, we removed that effect and re-ran the model. For GSI we used a Gaussian error distribution, which performed well because there were few values near 0% or 100%. For fecundity, we used a quasi-Poisson error distribution because data were counts and overdispersed. We did not include the covariate effect of embryo stage in models predicting fecundity, because preliminary analyses demonstrated that effect was non-significant (all $p > 0.07$) in each case, and past work shows partial atresia (a reduction in embryo number through embryonic development) is unsupported in mosquitofish [56]. For embryo diameter and embryo mass, we used a Gaussian error distribution, which performed well after $\log_{10}$ transformation of those response variables. We constructed these models in the R environment using the glm() function [57]. To obtain model coefficients and associated $p$-values, we used the summary() function. To obtain model $R^2$ values for the fecundity model, we used the package 'rsq' [58]. To approximately visualize models for GSI and $\log_{10}$ embryo size without the influence of embryo stage (the covariate), we obtained residuals from the OLS regression $trait \sim embryo\ stage$, and plotted predictions for the model $residuals \sim maternal\ length \times source\ temperature$.

## (e) F2 rearing

To assay juvenile growth rates, we reared newborn F2 fish across five treatment temperatures in two controlled environment rooms (TriMark R.W. Smith, San Diego, CA, USA) at UC Santa Cruz (electronic supplementary material, figure S2). The air temperature in one room was set to 23°C, and included tanks with treatment water temperatures 23, 29 and 32°C. The air temperature in the other room was set to 19°C, and included tanks with treatment water temperatures 19 and 26°C. The environmental settings in these two rooms were otherwise set to be identical, including a photoperiod set to 15 : 9 h daylight: dark. Tanks were 100 l plastic tubs ($91 \times 61 \times 20$ cm$^3$) filled with offgassed Santa Cruz city water. There were five replicate tanks per treatment temperature. In each room, tanks were randomly assigned a treatment temperature. Tanks with treatment temperatures above set air temperatures were warmed with submersible aquarium heaters. Water in all tanks was continuously homogenized with submersible water pumps (150 l per hour) to prevent thermal gradients within tanks. Tank water temperatures were monitored daily. We maintained water quality through siphoning of waste and 90% water changes biweekly. Fish were reared individually in cylindrical mesh containers with a Petri dish bottom and an open top (250 μm mesh, 7 cm diameter, 20 cm height). Fish containers were sunk into the water tanks, with the open top of container several centimetres above the water line, to prevent fish escape. Fish of a given source population were assigned temperature treatments sequentially as they were born, so that each population had approximately equal representation across the rearing temperatures. Fish of a given treatment temperature were then assigned one of the five replicate tubs sequentially as they were born, such that fish density differences among tubs were minimized through time. Fish were fed an excess of Frystartr food (Skretting, Stavanger, Norway) thrice daily. Growth was measured as the difference in total length at age 0 and at age 15 days. Lengths were measured from top-down photos taken with a scale bar (electronic supplementary material, figure S3), and analysed in ImageJ software [59].

## (f) F2 newborn size and juvenile growth

We tested for effects of source temperature on newborn size and on juvenile growth rates. For newborn size, we used the OLS regression: $log_{10}(length) \sim source\ temperature$. For juvenile growth rates, we used the OLS regression $growth \sim source\ temperature \times (rearing\ temperature)^2$. We included the second-order polynomial to allow for curvature in the effect of rearing temperature, as preliminary plots showed a curved pattern in growth across rearing temperatures. To do so, we used the poly() function in R, which creates orthogonal first- and second-order terms to allow interpretation of the significance of these coefficients separately. The interaction term was non-significant ($p = 0.332$), so we dropped that term from the model. Models were constructed using the lm() function. To obtain model coefficients and associated $p$-values, we used the summary() function. Diagnostics of the model predicting growth rates indicated possible violations of the homoscedasticity and normality assumptions (due to leptokurtosis). To evaluate whether this issue caused substantial problems for parameter estimates and statistical significance, we compared the model output with three 'robust' regression methods that deal with these issues. We did not observe substantive differences in the model output (electronic supplementary material, table S4), so here we report output from the simpler OLS regression. Finally, to test for differences in survival to age 15 days across source populations and rearing temperatures, we used a generalized linear model with a binomial error distribution. We used the model specification $survival \sim source\ temperature \times rearing\ temperature$, with rearing temperature treated as a factor. Data from this manuscript are available on Dryad [60].

# 3. Results

## (a) Reproductive traits

From field surveys in springtime, there was no support for an effect of site temperature on GSI or fecundity, though fecundity did increase with body size (figure 2$a$,$d$ and table 1). However, by summer, warmer-source populations showed a weaker increase in GSI and fecundity with increasing body size than did cooler-source populations (figure 2$b$,$e$), indicating that larger individuals performed relatively poorly at higher temperatures. Moreover, summertime samples indicated that individuals from warmer sites had higher fecundity at small sizes than did similarly sized fish from cooler sites (figure 2$e$). After common rearing, warmer-source populations expressed relatively high GSI and fecundity across all body sizes (figure 2$c$,$f$), supporting that recent evolution has led to an increase in overall reproductive effort in warmer-source populations.

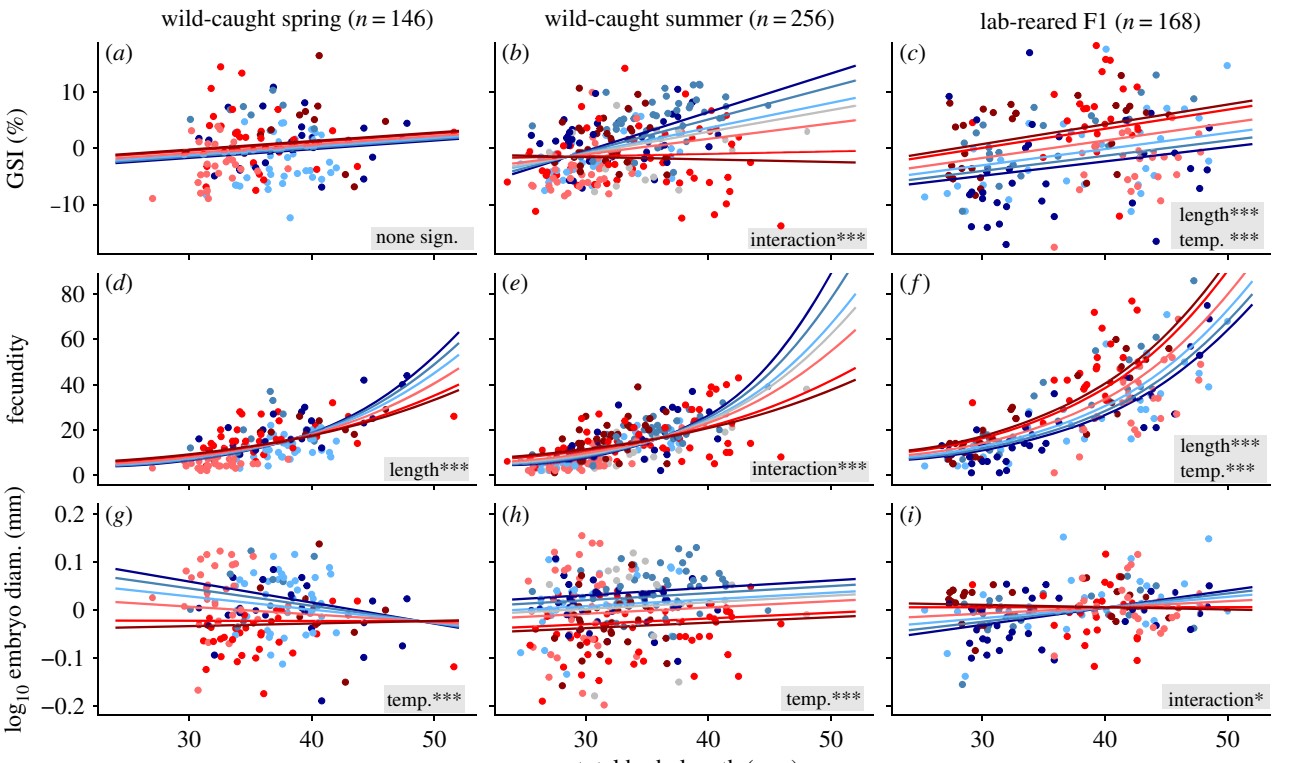

**Figure 2.** Effects of mosquitofish body length and source temperature on the focal life-history traits. To visualize these effects without the covariate effect of embryo stage (for GSI and embryo diameter only), we used residuals from the model *trait ∼ embryo stage*, and then plotted predictions for the model *trait residuals ∼ maternal fish length × temperature*. Interaction terms were always included for this graph, even if they were non-significant and removed for the final analysis in the main text (table 1). Points and prediction lines are coloured as in figure 1, with the 18.8°C source population labelled the darkest blue and the 33.3°C source population labelled the darkest red. Significant effects ($p < 0.001$***, 0.01** 0.05*) are included in the bottom-right corner of each panel (see table 1 for full statistical output). (Online version in colour.)

**Table 1.** Parameter estimates and significance ($p < 0.001$***, 0.01** 0.05*) from generalized linear models. Interaction effects removed due to non-significance ($p > 0.05$) are noted with 'RM'.

| trait response | sample | maternal length (mm) | site temperature (°C) | length × temperature | embryo stage [0,5] | intercept | adjusted $R^2$ |
|---|---|---|---|---|---|---|---|
| GSI (%) | spring | 0.1543 | 0.1072 | RM | 1.7234*** | 0.7209 | 0.2962 |
| | summer | 1.664*** | 1.4407** | −0.0509*** | 1.6470*** | −39.05*** | 0.4314 |
| | F1 | 0.3859*** | 0.4320*** | RM | 2.9438*** | −9.598* | 0.4878 |
| fecundity | spring | 0.0830*** | −0.0032 | RM | NA | −0.3393 | 0.3772 |
| | summer | 0.2007*** | 0.1525*** | −0.0042*** | NA | −4.4301*** | 0.4293 |
| | F1 | 0.0865*** | 0.0294*** | RM | NA | −0.7351* | 0.6029 |
| log embr. diam. (mm) | spring | −0.0019 | −0.0041*** | RM | 0.0539*** | 0.4180*** | 0.6867 |
| | summer | 0.0015 | −0.0050*** | RM | 0.0548*** | 0.2950*** | 0.7290 |
| | F1 | 0.0085** | 0.0103* | −0.0003* | 0.0380*** | −0.1040 | 0.6540 |
| log embr. mass (mg) | F1 | 0.0105*** | 0.0007 | RM | 0.0548*** | 0.0437 | 0.4229 |

## (b) Offspring size

In the wild, females from warmer-source populations had relatively small embryos, but common rearing showed weak support that this was caused by recent evolutionary divergence among populations. In both spring and summer, warmer-source populations had significantly smaller embryos (measured as diameter), and embryo size was unrelated to maternal size (figure 2g,h). After common rearing, we measured three metrics of offspring size. Analysis of embryo diameter suggested a weak interaction between maternal size and source temperature, such that cooler-source populations showed a slight increase in embryo size with increasing maternal size. This effect was reduced with increasing source temperature, as warmer-source populations

showed weak to no evidence of increased embryo diameter with increasing maternal size (figure 2i). Analysis of mean embryo mass showed no interaction or effect of source temperature, though embryo mass did increase with maternal size. Finally, analysis of F2 newborns showed no evidence that source temperature affected newborn length ($p = 0.292$; electronic supplementary material, figure S5). Thus, across these three metrics, there was weak evidence that source temperature caused substantial evolution of offspring size, and there was also weak evidence that maternal size substantially influences offspring size in this species.

## (c) F2 juvenile growth and survival

Across all rearing temperatures, second laboratory generation newborns from warmer-source populations exhibited slower growth rates than did newborn fish from cooler-source populations (figure 3a). We used model predictions for the coolest-source population reared at all temperatures and compared it with each source population theoretically reared at exactly its own source temperature. This exercise showed that ignoring evolution causes an overestimation of the acceleration of growth under warming (figure 3b). Moreover, with increasing magnitudes of warming, there is an increase in the importance of evolution as a proportion of growth (figure 3b). Finally, survival analysis on these F2 fish over their first 15 days of life showed approximately 75% overall survival and indicated no significant differences among source populations or rearing temperatures (all $p > 0.401$; electronic supplementary material, figure S6). Consequently, there is little evidence that selection during the second generation of rearing could have led to (a reduction of) observed differences in growth.

## 4. Discussion

We used recently established populations of mosquitofish (*Gambusia affinis*) across a unique geothermal temperature gradient from 19–33°C to test for effects of temperature on the recent evolution of size-related traits. Trait surveys of wild-caught fish indicated that higher temperatures reduce the benefit of large size. Specifically, warmer-source populations showed a weaker increase in GSI and fecundity with increasing body size compared to cooler-source populations. Higher temperatures were also associated with a substantial reduction in embryo size in the wild. After common-environment rearing to reveal recently evolved trait differences, warmer-source populations showed little difference in embryo or offspring size relative to cooler-source populations, indicating that the warming-induced reductions in offspring size observed in the wild may be caused by plasticity. However, after common rearing, warmer-source populations did exhibit a relative increase in reproductive effort and fecundity at small sizes and a decrease in juvenile growth rates. Altogether, these data are consistent with the hypothesis that warming disfavours large body size, leading to the evolution of increased reproduction at small sizes and to slowed somatic growth rates.

Our results support the notion that natural selection at warmer temperatures favours reduced size, and therefore that the plastic temperature–size rule may be adaptive. Specifically, our results show that warming alters fecundity selection, reducing the fecundity advantage of large body

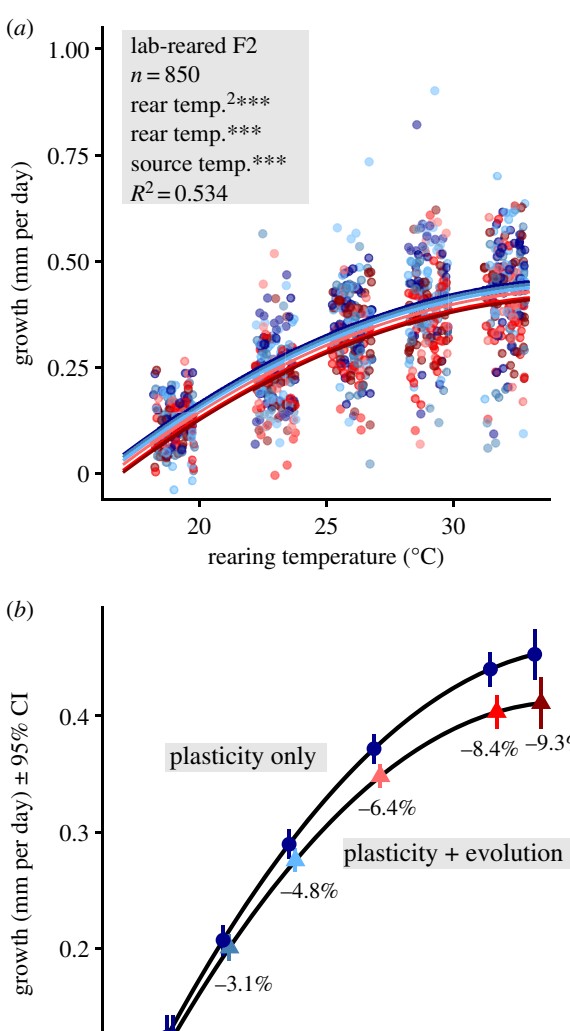

**Figure 3.** (a) Effects of source temperature on juvenile growth rates across the five rearing temperatures. Points and prediction lines are coloured as in figure 1, with the 18.8°C source population labelled the darkest blue and the 33.3°C source population labelled the darkest red. Points are jittered along the x-axis. Significant effects ($p < 0.001^{***}$, $0.01^{**}$ $0.05^{*}$) are highlighted in the top-left corner. (b) Predicted temperature dependence of growth rate for the 18.8°C source population (plasticity only) or for each source population reared at its own source temperature (plasticity + evolution). Shown are the percentage reductions in growth caused by evolution at each temperature relative to the 18.8°C population. Error bars represent 95% confidence intervals of the prediction. Note the difference in ranges on y-axes of each panel. (Online version in colour.)

size. A recent study showed a similar reduction in the fecundity advantage of large size in freshwater snails, and a simple life-history model suggested that this would favour the evolution of the temperature–size rule [37]. Here, in alignment with this life-history model, we show that populations from warmer sources have indeed recently evolved an increase in reproductive effort and fecundity at small body sizes. Interestingly, common-reared warmer-source populations also showed relatively high reproductive potential at large sizes, indicating that warmer-source populations are predisposed to greater reproduction throughout life. However, in nature, warmer-source populations were not able to sustain this high reproduction to later ages and larger sizes, probably

resulting from the stressors associated with living at higher temperatures and under natural conditions.

Although our data suggest that altered fecundity selection may contribute to the evolution of earlier and greater reproduction at a smaller size, our past work on mosquitofish populations in these geothermal springs also indicates that other forms of natural selection may be altered at warmer temperatures. In the wild, average female body sizes are smaller at warmer temperatures [50]. Female mosquitofish grow continuously throughout life, and their growth rates generally increase with temperature [51,52]. Therefore, it appears that mortality rates are higher for larger individuals in warmed systems, which life-history theory predicts can also lead to the evolution of the greater reproduction at a smaller size that we observed here [36]. Importantly, increased oxygen and resource stresses have been proposed as general mechanisms favouring reduced size under warming in aquatic environments [19]. Our data suggest multiple types of selection (mortality, fecundity) resulting from stressors in aquatic environments may contribute to similar patterns in body size evolution. These results may help to explain the widespread nature of the temperature–size rule in aquatic taxa [13], and may explain why terrestrial organisms show less consistent patterns of selection [61]. Looking forward, a more precise understanding of the drivers underlying warming-induced evolution could be achieved by parsing the relative contributions of altered mortality versus fecundity selection for different species and in different environments.

In our study, common-reared warmer-source populations displayed slower juvenile growth rates across a wide range of rearing temperatures. This countergradient variation in growth rates has been found along latitudinal temperature gradients in many other fish species [23]. In those cases, it had not been clear whether temperature was the driver of countergradient variation in growth rates, because factors like seasonality of light and resource availability also vary systematically along those thermal gradients. In our study system, the thermal gradient does not appear to be strongly confounded with at least two metrics of basal resource availability—nutrients and chlorophyll $a$ (electronic supplementary material, table S1). More importantly, a large component of the diet of mosquitofish in these springs comes from allochthonous, or outside, sources (i.e. terrestrial insects), effectively decoupling local resource production from resource availability (E.R.M. 2016, personal observation). In addition, the seasonality of photoperiod was not likely to covary with temperature, because all sites are in close proximity (figure 1b). Thus, it seems likely here that temperature itself led to the evolution of countergradient variation in growth. It may be that this reduction in growth arises as a correlated response to selection on metabolic traits, but past work in other systems shows mixed evidence for temperature-induced countergradient variation in metabolism (e.g. [62,63]). Alternatively, warming-induced evolution of reduced growth early in life may result from a trade-off between growth rate and resistance to oxidative stress [64]. The elevated metabolism experienced by animals under warming may increase their exposure to harmful oxygen species [65], causing a shift along this trade-off towards slower growth. Regardless of the reasons for this reduction in growth, it is clear from our results that reduced individual growth rates can evolve quickly in response to temperature.

This reduction in early growth, coupled with a likely decrease in somatic growth after maturity, is likely to contribute to reduced body size distributions at warmer temperatures in this species. Because countergradient variation in growth is common across a variety of taxa, and widely observed in fishes, these altered growth and development schedules may be important for predicting fisheries sustainability and yields in a warming world.

Our data emphasize a role for both plasticity and evolution in contributing to reduced body size under warming. Field surveys indicated a strong reduction in offspring size at higher temperatures, that common rearing suggested was due to plasticity. Laboratory rearing in the sister species G. holbrooki also shows that higher rearing temperatures cause a plastic decline in size at maturity [66]. Therefore, mosquitofish, like many taxa, support the temperature–size rule of reduced stage-specific body sizes at higher rearing temperatures due to plasticity. However, our data also support a role for rapid evolution in contributing to these plastic size declines. First, evolution caused a reduction of juvenile growth rates. Second, evolution caused an increase in reproductive effort at small sizes. This increase in reproductive effort is likely to exacerbate the reduction in somatic growth rates after maturity. Altogether, this plasticity and evolution, coupled with the demographic effects of likely increases in mortality rates under warming, suggest that multiple processes may combine to produce a substantial decline in population body size distributions under warming [67]. However, to extend the trait evolution found here to population-level outcomes, future work should aim to (1) understand the combined effects of evolution and plasticity, including potential transgenerational plasticity not accounted for here [68], and (2) evaluate the role of warming for the evolution of body size, growth, and reproductive success in males as well.

Overall, this study provides evidence that evolutionary adaptation to temperature itself is likely to contribute to reduced body size and growth rates over short time scales, compounding size reductions caused by other mechanisms. Body size is a key functional trait [1–6], and for fishes and other taxa, it is of key economic importance [69]. Although it is yet unknown whether the degree of evolution here is sufficient to significantly alter ecological dynamics, it is clear that this evolution can happen over the short time scales required to potentially affect these outcomes in the near-term future. Indeed, a young but growing literature suggests that rapid evolution and body size changes may significantly mediate the ecological consequences of warming [15,16,18,70–72]. Thus, if we aim to forecast the ecological consequences of warming for populations, ecosystems and society, we may need to incorporate body size and growth evolution into these models.

Ethics. Collections were approved by our institutional animal ethics committee (UCSC protocols PALKE-1311 and PALKE-1801) and the local wildlife agency (CADFW permit SC-12752).

Data accessibility. Data are available on the Dryad Digital Repository: https://doi.org/10.5061/dryad.cc2fqz63k [60].

Authors' contributions. D.C.F. led the study design, with contributions from E.R.M., M.T.K., K.S.S. and E.P.P. D.C.F. led collection of field data and wild fish, and performed common rearing (with help from F.J.A. and J.N.B.). D.C.F. developed the dissection protocol. A.N.H. led wild-caught trait measurements. D.A.A. led F1 trait measurements. D.C.F. and F.J.A. led F2 trait measurements. D.C.F. performed the statistical analyses. D.C.F. wrote the first draft of the manuscript, which was edited by all authors.

**Competing interests.** We declare we have no competing interests.

**Funding.** This research was supported by our affiliated institutions, the US National Science Foundation (a GRF to D.C.F., DEB 1457333 to E.P.P. and DEB 1457112 to M.T.K.), and the Royal Society of New Zealand Marsden Fund (16-UOA-023) to K.S.S., E.P.P. and M.T.K. E.P.P. received partial support from the Cooperative Institute for Marine Ecosystems and Climate.

**Acknowledgements.** We thank S. Parmenter, P. Pister and C. Wickham for introducing us to these geothermal springs populations. For technical assistance, we thank T. Apgar, R. Franks and J. Velzy. For help with field collections, we thank K. Morrison, J. Penfield, S. Munch and B. Wasserman. For help with dissections and trait measurements, we thank E. Portillo, T. Amarnath, S. Bell, R. Robinson, E. Glensky, M. Haptonstall and M. De Aquino. For help with fish care during common rearing, we thank D. Ruiz, D. Weiler and T. Vance.

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
