## [Reviewer comments · Proceedings of the Royal Society B: Biological Sciences]

Review History

RSPB-2020-0608.R0 (Original submission)

Review form: Reviewer 1

Recommendation

Accept with minor revision (please list in comments)

Scientific importance: Is the manuscript an original and important contribution to its field?

Good

General interest: Is the paper of sufficient general interest?

Good

Quality of the paper: Is the overall quality of the paper suitable?

Good

Is the length of the paper justified?

Yes

Should the paper be seen by a specialist statistical reviewer?

No

Do you have any concerns about statistical analyses in this paper? If so, please specify them explicitly in your report.

No

It is a condition of publication that authors make their supporting data, code and materials available - either as supplementary material or hosted in an external repository. Please rate, if applicable, the supporting data on the following criteria.

Is it accessible?

Yes

Is it clear?

Yes

Is it adequate?

Yes

Do you have any ethical concerns with this paper?

No

Comments to the Author

Fryxell et al., conducted a nice study in which they sought to determine whether there was evidence of an evolutionary response of reproductive traits (fecundity, GSI and embryo size) and growth rates to variation in habitat temperature. To achieve this, they examined wild caught individuals from sites with distinct temperature regimes and paired these data with a common garden rearing experiment along with temperature treatments. I think researchers interested in the effects of temperature and evolution would be interested in this paper. In general, I thought the study was well designed and appropriately analyzed. I only have a one main concern regarding the clarity of the discussion and patterns of F2 body size. However, I thought the lack of inclusion of males should be discussed. I understand the reasoning, but does this limit the generality of the results of this study?

A key statement is in lines 337 – 338 is about how large size is unlikely “achieved” at warm sites. This should be discussed more explicitly. You touch on that it is likely due to restricted growth at high temps, but it could also be due to selective mortality of larger sized individuals. In figure 3 the fastest growth rates were at the highest temperatures regardless of source population, this to me suggests that there isn’t necessarily a large inherent cost to growth at a high temperature alone. Can you show a plot of final F2 body size distribution by temperature treatment? Is it the case that in the lab at high temps you still have large individuals? This would perhaps suggest that there is actually selection against large size in the wild independent (or partially) of growth. Inclusion of this data and discussion of the pattern may help to disentangle the effects of plasticity and different agents of selection.

Also was there a relationship between survival, body size and temperature in the lab? If not, it may suggest some ecological agent of selection against large body size in the wild.

In general, these distinct mechanisms of selection (growth restriction and differential mortality) should be more clearly presented and the support for the contribution of each mechanism to explaining the observed patterns should be outlined. I think the ideas are there, but the organization of thoughts is such that there is frequent switching of topics which makes the logic difficult to follow.

It would be useful to have a plot showing the size distributions from each site in the wild. It is hard to see the individual distributions of size in figure 2. I think this is important as it doesn’t seem like there are nearly as many large fish in the wild sample at warm sites and that influences the relationship of course (and this is later touched on in the discussion).

How could sexual dimorphism in growth pattern affect the results and generalization of results beyond this species? Is it common for adults to keep growing beyond point of maturity in other taxa? Could sexual dimorphism in growth patterns constrain evolution in this species. I would suggest this should be at least briefly discussed.

Minor comments:

Line 67 confounding factors? This sentence could be combined with the next for an easier read.

Line 276-277 is confusing as written.

Line 329, the word "be" is missing between "may" and "adaptive".

In figure 1, panel c it is very hard to see the individual sites. Is it possible to average the data or plot it another way so the variance is clearer?

Review form: Reviewer 2

Recommendation

Accept with minor revision (please list in comments)

Scientific importance: Is the manuscript an original and important contribution to its field?

Excellent

General interest: Is the paper of sufficient general interest?

Excellent

Quality of the paper: Is the overall quality of the paper suitable?

Excellent

Is the length of the paper justified?

Yes

Should the paper be seen by a specialist statistical reviewer?

No

Do you have any concerns about statistical analyses in this paper? If so, please specify them explicitly in your report.

No

It is a condition of publication that authors make their supporting data, code and materials available - either as supplementary material or hosted in an external repository. Please rate, if applicable, the supporting data on the following criteria.

Is it accessible?

N/A

Is it clear?

N/A

Is it adequate?

N/A

Do you have any ethical concerns with this paper?

No

Comments to the Author

This is a very well written paper. It is clear the authors have proofread the final draft and removed typographical and obvious grammatical errors before submitting. These days I rarely see this, even in the final published version of manuscripts and greatly appreciate your having done so.

I found the paper well written and well reasoned. I have only one minor issue with the work presented here, that reproductive data were not taken on the F2 lab-reared fish. Wild-caught fish obviously retain residual effects of the native environment. I'm glad you used fish born >1 month after the wild fish were caught as that does eliminate native environmental effects on the F1 fish. However, there could still be maternal effects that can obscure genetic differences. I am not aware that anyone has shown strong maternal effects on these characters in *Gambusia* so I do not think this is a major concern. However, I would like to see this acknowledged in the Discussion.

I also found the reference to "small-sized taxa" on line 72 a little confusing/misleading. Conover and Present (your reference 41) did precisely this with Atlantic Silverside which are not really small-sized, at least in comparison to *Gambusia affinis*.

I do like the comparison of plastic vs. genetic effects on growth in the F2 fish summarized in Fig. 3b. This is obviously what one expects from countergradient variation and I thought this must have been shown like this in the past. However, after a quick look through my papers on this subject I could not find an example where this pattern is presented in quite the same way.

Decision letter (RSPB-2020-0608.R0)

20-Apr-2020

Dear Dr Fryxell:

Your manuscript has now been peer reviewed and the reviews have been assessed by an Associate Editor. The reviewers' comments (not including confidential comments to the Editor) and the comments from the Associate Editor are included at the end of this email for your reference. As you will see, the reviewers and the Editors have raised some concerns with your manuscript and we would like to invite you to revise your manuscript to address them. I would like to echo the sentiments of the board member, and that while the revisions may be categorised as minor, they do represent substantive issues that require your careful consideration. As indicated below, the invitation to revise your manuscript, does not guarantee eventual acceptance, and the nature of your responses to the issues raised, will be vital, in terms of any decision about forward processing of your manuscript.

Research ethics:

Use of animals and field studies:

Please submit a copy of your revised paper within three weeks. If we do not hear from you within this time your manuscript will be rejected. If you are unable to meet this deadline please let us know as soon as possible, as we may be able to grant a short extension.

Best wishes,
Professor Gary Carvalho
mailto:proceedingsb@royalsociety.org

Associate Editor

Comments to Author:

Two reviewers gave the paper favorable reviews and suggest minor revisions. Though minor, both reviewers suggest substantive changes to data interpretation, so please give their suggestions careful attention.

Reviewer(s)' Comments to Author:

Referee: 1

Comments to the Author(s)

Fryxell et al., conducted a nice study in which they sought to determine whether there was evidence of an evolutionary response of reproductive traits (fecundity, GSI and embryo size) and growth rates to variation in habitat temperature. To achieve this, they examined wild caught individuals from sites with distinct temperature regimes and paired these data with a common garden rearing experiment along with temperature treatments. I think researchers interested in the effects of temperature and evolution would be interested in this paper. In general, I thought the study was well designed and appropriately analyzed. I only have a one main concern regarding the clarity of the discussion and patterns of F2 body size. However, I thought the lack of inclusion of males should be discussed. I understand the reasoning, but does this limit the generality of the results of this study?

A key statement in lines 337 – 338 is about how large size is unlikely “achieved” at warm sites. This should be discussed more explicitly. You touch on that it is likely due to restricted growth at high temps, but it could also be due to selective mortality of larger sized individuals. In figure 3 the fastest growth rates were at the highest temperatures regardless of source population, this to me suggests that there isn't necessarily a large inherent cost to growth at a high temperature alone. Can you show a plot of final F2 body size distribution by temperature treatment? Is it the case that in the lab at high temps you still have large individuals? This would perhaps suggest that there is actually selection against large size in the wild independent (or partially) of growth. Inclusion of this data and discussion of the pattern may help to disentangle the effects of plasticity and different agents of selection.

Also was there a relationship between survival, body size and temperature in the lab? If not, it may suggest some ecological agent of selection against large body size in the wild.

In general, these distinct mechanisms of selection (growth restriction and differential mortality) should be more clearly presented and the support for the contribution of each mechanism to

explaining the observed patterns should be outlined. I think the ideas are there, but the organization of thoughts is such that there is frequent switching of topics which makes the logic difficult to follow.

It would be useful to have a plot showing the size distributions from each site in the wild. It is hard to see the individual distributions of size in figure 2. I think this is important as it doesn't seem like there are nearly as many large fish in the wild sample at warm sites and that influences the relationship of course (and this is later touched on in the discussion).

How could sexual dimorphism in growth pattern affect the results and generalization of results beyond this species? Is it common for adults to keep growing beyond point of maturity in other taxa? Could sexual dimorphism in growth patterns constrain evolution in this species. I would suggest this should be at least briefly discussed.

Minor comments:

Line 67 confounding factors? This sentence could be combined with the next for an easier read.

Line 276-277 is confusing as written.

Line 329, the word "be" is missing between "may" and "adaptive".

In figure 1, panel c it is very hard to see the individual sites. Is it possible to average the data or plot it another way so the variance is clearer?

Referee: 2

Comments to the Author(s)

This is a very well written paper. It is clear the authors have proofread the final draft and removed typographical and obvious grammatical errors before submitting. These days I rarely see this, even in the final published version of manuscripts and greatly appreciate your having done so.

I found the paper well written and well reasoned. I have only one minor issue with the work presented here, that reproductive data were not taken on the F2 lab-reared fish. Wild-caught fish obviously retain residual effects of the native environment. I'm glad you used fish born >1 month after the wild fish were caught as that does eliminate native environmental effects on the F1 fish. However, there could still be maternal effects that can obscure genetic differences. I am not aware that anyone has shown strong maternal effects on these characters in *Gambusia* so I do not think this is a major concern. However, I would like to see this acknowledged in the Discussion.

I also found the reference to "small-sized taxa" on line 72 a little confusing/misleading. Conover and Present (your reference 41) did precisely this with Atlantic Silverside which are not really small-sized, at least in comparison to *Gambusia affinis*.

I do like the comparison of plastic vs. genetic effects on growth in the F2 fish summarized in Fig. 3b. This is obviously what one expects from countergradient variation and I thought this must have been shown like this in the past. However, after a quick look through my papers on this subject I could not find an example where this pattern is presented in quite the same way.

Author's Response to Decision Letter for (RSPB-2020-0608.R0)

See Appendix A.

RSPB-2020-0608.R1 (Revision)

Review form: Reviewer 1

Recommendation

Accept as is

Scientific importance: Is the manuscript an original and important contribution to its field?

Good

General interest: Is the paper of sufficient general interest?

Good

Quality of the paper: Is the overall quality of the paper suitable?

Good

Is the length of the paper justified?

Yes

Should the paper be seen by a specialist statistical reviewer?

No

Do you have any concerns about statistical analyses in this paper? If so, please specify them explicitly in your report.

No

It is a condition of publication that authors make their supporting data, code and materials available - either as supplementary material or hosted in an external repository. Please rate, if applicable, the supporting data on the following criteria.

Is it accessible?

Yes

Is it clear?

Yes

Is it adequate?

Yes

Do you have any ethical concerns with this paper?

No

Comments to the Author

I am satisfied with the revisions of the manuscript and do not suggest any further revisions.

Decision letter (RSPB-2020-0608.R1)

06-May-2020

Dear Dr Fryxell

I am pleased to inform you that your manuscript entitled "Recent warming reduces the reproductive advantage of large size and contributes to evolutionary downsizing in nature" has been accepted for publication in Proceedings B.

Open Access

Paper charges

Sincerely,

Professor Gary Carvalho

Associate Editor:

Board Member: 1

Comments to Author:

(There are no comments.)

Board Member: 2

Comments to Author:

(There are no comments.)

Appendix A

Dear PRSB Editorial Team,

Please find our response to reviewer comments in this document. Below these responses, we include a tracked changes version of the manuscript.

We have now carefully considered and addressed each of the reviewer suggestions in bold type below. We were able to accommodate many of the suggested additions and clarifications. In particular, we now (1) justify our focus on female fish and we acknowledge that future work would benefit from studying males, (2) we clarify what was meant that the large sizes “achieved” in the lab are unlikely to be achieved at warmer sites in the wild, and (3) we acknowledge the unknown role of transgenerational plasticity given our study design. Reviewer 1 suggested some additions that could not be made because they require data that we do not have. Nevertheless, we carefully considered all suggestions and we include our responses to each suggestion below. Please note that line numbers provided in our response refer to the updated manuscript file with changes accepted, rather than the line numbers of the manuscript viewed with changes tracked.

We appreciate the time and effort put in by your editorial team and the reviewers. We hope you find this revision satisfactory for publication in PRSB.

Sincerely,

David Fryxell and coauthors

Reviewer comments and responses

Associate Editor

Comments to Author:

Two reviewers gave the paper favorable reviews and suggest minor revisions. Though minor, both reviewers suggest substantive changes to data interpretation, so please give their suggestions careful attention.

Reviewer(s)' Comments to Author:

Referee: 1

Comments to the Author(s)

Fryxell et al., conducted a nice study in which they sought to determine whether there was evidence of an evolutionary response of reproductive traits (fecundity, GSI and embryo size) and growth rates to variation in habitat temperature. To achieve this, they examined wild caught individuals from sites with distinct temperature regimes and paired these data with a common garden rearing experiment along with temperature treatments. I think researchers interested in the effects of temperature and evolution would be interested in this paper. In general, I thought the study was well designed and appropriately analyzed.

Thank you.

I only have a one main concern regarding the clarity of the discussion and patterns of F2 body size. However, I thought the lack of inclusion of males should be discussed. I understand the reasoning, but does this limit the generality of the results of this study?

We focus on females because our hypothesis surrounded fecundity selection. A further advantage of focusing on females for some readers may be that females, unlike male *Gambusia*, show indeterminate growth, which is common for both sexes in many fish species (now stated and with a new reference, Lines 133-135). In any case, our hypotheses do not hinge on whether growth is indeterminate, so the mechanisms discussed here should apply to all animals. Nonetheless, we acknowledge that an understanding of male evolution is important to scaling up the patterns in female evolution found here to population-level outcomes, so we now point this out in the Discussion (Lines 401-402).

A key statement in lines 337 – 338 is about how large size is unlikely “achieved” at warm sites. This should be discussed more explicitly. You touch on that it is likely due to restricted growth at high temps, but it could also be due to selective mortality of larger sized individuals. In figure 3 the fastest growth rates were at the highest temperatures regardless of source population, this to me suggests that there isn’t necessarily a large inherent cost to growth at a high temperature alone.

We have now rewritten this section to clarify 1) what we meant when saying large body sizes reached in the lab are not commonly achieved in the wild (Lines 337-342) and 2) that reduced size may be favored by either mortality or fecundity selection (what we think you are referring to as ‘growth restriction,’ a by-product of warming-induced fecundity selection causing greater reproductive effort in mature females) (Lines 343-346, 356-359). Notably, the growth rates in Fig 3 are for juveniles, before allocation towards reproduction, so these data do not directly address the fecundity selection hypothesis.

Can you show a plot of final F2 body size distribution by temperature treatment? Is it the case that in the lab at high temps you still have large individuals? This would perhaps suggest that there is actually selection against large size in the wild independent (or partially) of growth. Inclusion of this data and discussion of the pattern may help to disentangle the effects of plasticity and different agents of selection.

F2 common reared fish were reared to age 15 days, at which point they were still immature and small. So, unfortunately, we do not have data on the maximum achievable body sizes for individuals reared at different temperatures.

Also was there a relationship between survival, body size and temperature in the lab? If not, it may suggest some ecological agent of selection against large body size in the wild.

There was no effect of rearing temperature on F2 newborn survival (Lines 309-313). For F0 and F1 fish, we were not able to test for survival at different rearing temperatures because they were all reared at the same temperature. We were also unable to test for different survival by body size for these fish; F0 and F1 rearing was done at the population level, so we did not track individual fish. Nonetheless, we do appreciate your comment and we acknowledge the point that mortality in nature is an important unmeasured mechanism that could also contribute to the patterns of evolution. We

added a sentence to emphasize that this should be a future line of research (Lines 356-359).

In general, these distinct mechanisms of selection (growth restriction and differential mortality) should be more clearly presented and the support for the contribution of each mechanism to explaining the observed patterns should be outlined. I think the ideas are there, but the organization of thoughts is such that there is frequent switching of topics which makes the logic difficult to follow.

We have made edits that we hope clarify these mechanisms (Lines 330-359). Specifically, we now discuss each mechanism in its own paragraph (previously we discussed both mechanisms in one paragraph). Further, we add a line that emphasizes the need to parse these mechanisms in future research (Lines 356-359).

It would be useful to have a plot showing the size distributions from each site in the wild. It is hard to see the individual distributions of size in figure 2. I think this is important as it doesn't seem like there are nearly as many large fish in the wild sample at warm sites and that influences the relationship of course (and this is later touched on in the discussion).

We prefer not to include this graph because we did not sample in a way that would provide accurate size distributions in the wild. Rather we sampled to collect fish from a similar range in body size across all sites (stated on Line 195) and using various capture methods to different degrees at each site (targeted hand netting which may be size-biased versus seines, stated Line 152-153). However, in our previously published work in this system we had done standardized seine hauls at all sites to estimate size distributions (Moffett et al. 2018). We cite this paper to justify our statement that body size distributions become smaller at higher temperatures (Lines 110, 347).

How could sexual dimorphism in growth pattern affect the results and generalization of results beyond this species? Is it common for adults to keep growing beyond point of maturity in other taxa? Could sexual dimorphism in growth patterns constrain evolution in this species. I would suggest this should be at least briefly discussed.

Indeterminate growth is the norm in fishes, so female mosquitofish are not unique in this regard (now stated in Lines 133-135). Nevertheless, our hypothesis does not only apply to species or sexes with indeterminate growth.

Minor comments:

Line 67 confounding factors? This sentence could be combined with the next for an easier read.

We changed the wording to “confounding factors” (Lines 66-67).

Line 276-277 is confusing as written.

Rewritten to increase clarity (Lines 279-282).

Line 329, the word “be” is missing between “may” and “adaptive”.

Added "be."

In figure 1, panel c it is very hard to see the individual sites. Is it possible to average the data or plot it another way so the variance is clearer?

The figure is much clearer now that it is uploaded separately at full size and not embedded within the .docx document. Nevertheless, we also calculate the temperature variance and report it in Table S1.

Referee: 2

Comments to the Author(s)

This is a very well written paper. It is clear the authors have proofread the final draft and removed typographical and obvious grammatical errors before submitting. These days I rarely see this, even in the final published version of manuscripts and greatly appreciate your having done so.

Thank you.

I found the paper well written and well reasoned. I have only one minor issue with the work presented here, that reproductive data were not taken on the F2 lab-reared fish. Wild-caught fish obviously retain residual effects of the native environment. I'm glad you used fish born >1 month after the wild fish were caught as that does eliminate native environmental effects on the F1 fish. However, there could still be maternal effects that can obscure genetic differences. I am not aware that anyone has shown strong maternal effects on these characters in *Gambusia* so I do not think this is a major concern. However, I would like to see this acknowledged in the Discussion.

Fair point. We took the approach of measuring reproductive traits on F1 fish and not F2 fish due to logistical constraints. Rearing to maturity takes a long time, and doing so for two full generations was not feasible under our research timeline. We now address the possibility that our design may not account for some forms of transgenerational plasticity in the Discussion and we recommend future directions of research along these lines (Lines 398-402).

I also found the reference to "small-sized taxa" on line 72 a little confusing/misleading. Conover and Present (your reference 41) did precisely this with Atlantic Silverside which are not really small-sized, at least in comparison to *Gambusia affinis*.

We now clarify by stating that the experimental evolution studies we refer to are those that have manipulated temperature as an agent of selection. This type of work has previously been restricted to smaller taxa like phytoplankton, zooplankton, and small terrestrial arthropods (now stated, with appropriate references, on Lines 70-73). The Conover study you reference performed common rearing of fish from different latitudes, but they did not perform experimental evolution. They also did not attribute the countergradient pattern to temperature per se. Rather, they attributed it to the length of

the growing season (the title of that paper is “Countergradient variation in growth rate: compensation for length of the growing season among Atlantic silversides from different latitudes”).

I do like the comparison of plastic vs. genetic effects on growth in the F2 fish summarized in Fig. 3b. This is obviously what one expects from countergradient variation and I thought this must have been shown like this in the past. However, after a quick look through my papers on this subject I could not find an example where this pattern is presented in quite the same way.

Thank you. This approach may not have been taken in the past because most past work (including all cases with fishes) is less able to attribute temperature per se (versus seasonality or latitude) as the driver of countergradient evolution.